# An Ontological Framework to Facilitate Early Detection of ‘Radicalization’ (OFEDR)—A Three World Perspective

**DOI:** 10.3390/jimaging7030060

**Published:** 2021-03-22

**Authors:** Linda Wendelberg

**Affiliations:** Department of Information Security and Communication Technology, Faculty of Information Technology and Electrical Engineering, Norwegian University of Science and Technology (NTNU), 2815 Gjøvik, Norway; wendelberg@yahoo.no; Tel.: +47-48-220-867

**Keywords:** ontology, existential anxiety, radicalization, brain, behavior, biochemistry, individual characteristics, crime profiles, protégé, SPARQL

## Abstract

This paper presents an ontology that involves using information from various sources from different disciplines and combining it in order to predict whether a given person is in a radicalization process. The purpose of the ontology is to improve the early detection of radicalization in persons, thereby contributing to increasing the extent to which the unwanted escalation of radicalization processes can be prevented. The ontology combines findings related to existential anxiety that are related to political radicalization with well-known criminal profiles or radicalization findings. The software Protégé, delivered by the technical field at Stanford University, including the SPARQL tab, is used to develop and test the ontology. The testing, which involved five models, showed that the ontology could detect individuals according to “risk profiles” for subjects based on existential anxiety. SPARQL queries showed an average detection probability of 5% including only a risk population and 2% on a whole test population. Testing by using machine learning algorithms proved that inclusion of less than four variables in each model produced unreliable results. This suggest that the Ontology Framework to Facilitate Early Detection of ‘Radicalization’ (OFEDR) ontology risk model should consist of at least four variables to reach a certain level of reliability. Analysis shows that use of a probability based on an estimated risk of terrorism may produce a gap between the number of subjects who actually have early signs of radicalization and those found by using probability estimates for extremely rare events. It is reasoned that an ontology exists as a world three object in the real world.

## 1. Introduction

In the fall of 2019, Philip Manshaus killed his sister and, on the same day, attacked a mosque in Bærum, Norway [1]. Identifying potential terrorists, including the early start of radicalization processes, has proved to be difficult. In addition, there is a lack of reliable indicators of change in individuals related to the escalation of radicalization processes [2]. Moreover, knowledge about radicalization indicators is spread across different disciplines without these subject areas communicating and collaborating with one another. One example is neurophysiological and neuropsychological indicators that are most frequently developed by medicine and psychology. Another example is indicators related to terrorist acts that are developed by law enforcement agencies (LEA) and the defense sector. These disciplines operate in real life as separate silos that are largely prevented from cooperating because of basic regulations, such as confidentiality and organization. Another issue is that many institutions, such as schools, work-life, universities, parents, hospitals, etc., require knowledge and indicators that are reserved for the police and defense. Nonetheless, early signs of radicalization or escalation of radicalization may be more easily detected in real life, outside agencies which deal with security. LEAs are often dependent on information from individuals and other institutions and organizations. One possible reason for the challenge of identifying radicalization may be sharp dividing lines between disciplines. Ontologies can help due to the possibility of using data sources from different disciplines, which can be considered a possible strategy or method to circumvent challenges related to the segregation of knowledge. Nevertheless, the development of ontologies and even the definition of an ontology is no simple, straightforward issue, especially not when one needs to cross professional boundaries.

### 1.1. Terror Management and Terrorism

The Terror Management Theory (TMT) states that existential anxiety (EA) is the activation of a mental defense aimed at protecting the individual from the fear of death [3]. EA is a very common condition that can be triggered by, for example, cemeteries and other death reminders. Nevertheless, this condition is related to political radicalization even if the physical link between this condition and terrorist acts is still lacking [4,5]. Terrorist acts are rare events and account for a relatively small amount of deaths worldwide [6]. Considering the global population, the chance that one individual will be a terrorist is very low, at 1.1 $\times \;10^{-6}$ (global attacks in 2019/world population in 2019) [7,8]. This suggest that if EA is related to radicalization with a violent end, then a frequently experienced condition (EA) can be related to a rarely occurring event (terrorism). Research suggests that most papers do not provide an explicit definition of extreme events [9]. Moreover, there is no unified definition regarding the content or distribution of an extreme event. In analyses, extreme events are frequently separated from everything else, though we know from cases in the world that a signal and often many signals were sent into the environment in advance of the incident. This work challenges the concept of extreme events by applying a condition that frequently occurs in order to hypothetically predict the risk of something that happens very rarely, because it is a fact that society made observations before an extreme event that were not taken seriously. Could it be that it is better to look for small signs instead of investigating the probability of extreme events? EA concerns meaning and existence in the world when we face reminders of our own death.

### 1.2. Ontology and Popper’s Three Worlds

Ontologies belong to metaphysics (a branch of philosophy), which investigates the nature of being (ontology) and reality. Ontologies are linked to existence as opposed to epistemology, which could also be labeled knowledge development. The dividing lines between ontology and knowledge development can be difficult to define. For example, ontology asks whether there is a God (what), metaphysics asks what rules govern God’s actions (how), and epistemology asks if humans can know whether God exists. Ontologies are about existence and reality. In 1978, Karl Popper presented a lecture where he described an ontological “system” without using the word ontology [10]. How can we define existence/being/reality? The philosopher and science theorist Popper described “three ontological worlds”, or domains referred to as three kinds of realities. These “realities” can be used to understand the content of being/existence (see Table 1 for a description of Popper’s three worlds).

The three worlds can be considered to describe the set of things/objects on which ontologies rely. Moreover, these sets of things can give many combinations depending on their relationship (world 3), which gives associations to what we today call triplets in ontologies. Triplets are a set of entities that classify semantic data statements into subject-predicate-object expressions [11]. For example, John Doe (subject world one) has smaller (predicate world three) pupil diameter (world one), or John Doe (subject world one) has radical (predicate world three) values/beliefs (world two). An ontology may consist of world one to three objects. By including world three objects, we can say that an ontology is also a system because world three objects can have a logical relationship and may affect the world one objects.

### 1.3. Terror Management—Three World Ontology

In an ontology, a world three object can be said to represent an object with a property (object/data). Karl Popper suggests that whatever exists is whatever may have a causal effect on physical things [10]. According to his theory, EA should exist because the condition can change physiological expressions. Moreover, the TMT should exist because it provides a “plan” to detect EA. He also suggests that it is world three objects that gives world two objects the power to change world one [10] (p. 156). Individuals in an ontology are world one objects. Classes may represent things that exist in worlds one to three. In summary, these characteristics make an ontology a system, as wisely described by Ervin László’s idea of the existence of a system that captures patterns across specialized disciplines that are organized as a whole [12]. More importantly, an ontology in itself can be described as an “integrated whole of its subsidiary components” because entities in ontologies have different kinds of relationships and causalities [12] (p. 14). EA is characterized by a change in values and beliefs—a world two object—, though, when we ask whether this change is true or not, the object changes to a world three object.

### 1.4. Ontology Definition

An ontology can be defined as a “common vocabulary for researchers who need to share information in a domain” [13] (p. 1). Ontologies are considered to be at the semantic level (world three) rather than at the logical or physical level as found in databases [14]. The field of artificial intelligence (AI) adopted the term ontology from the field of philosophy [14,15]. Today, the semantic web, to a large degree, relies on formal ontologies to structure and transport interpretive understandings [16]. Because of the massive development in the web environment (unstructured knowledge), ontologies are included as an important part of the AI field. Ontologies facilitate the handling of unstructured knowledge and contribute to common discourse and development based on reuse and shared understanding (common language). An ontology provides a language for communicating across scientific disciplines [14]. The common language Web Ontology Language (OWL) is built on the World Wide Web Consortium’s (W3C) XML standard for objects called the Resource Description Framework (RDF) [11].

The use of a common language between disciplines is important concerning the present work, as well as kinds of work that include different levels and dimensions of a “thing”. The “thing” (a thing = something that exists) in the present work can be described according to the three worlds described by Karl Popper.

### 1.5. State of the Art

The TMT is a theory that can be classified as belonging to Popper’s third world. The findings related to the theory can, for example, be classified as human observations (world two) or physical measures (world one), depending on the types of findings. EA is related to the “radicalization” of already existing values, beliefs, and behaviors, including political radicalization [4]. In addition to political choices/selection, a meta-study by Burke and colleagues included some studies measuring support for same-sex partner benefits, support for a health care reform, and support for national military force. EA is, in addition to political radicalization, documented to be a related change in sexual behavior or mating behavior, and objectification [4,17,18,19]. Individual characteristics moderate EA (I) [20,21]. In summary, a wide range of studies document that EA contributes to change in, for example, behavior (B), brain processing (B), and biochemistry (B) [3,22]. However, the existence of a physical link between EA and violent radicalization is still lacking. How is it possible to argue that there is a relationship between EA and criminal profiles (CPs)? One presupposition for the present work is that research findings in the field of radicalization/extremism or existing criminal profiles may cover attributes (in individuals) that overlap with research findings regarding changes in attributes (in individuals) related to EA. The field of radicalization and extremism is under rapid development, and, still today, no universally agreed-upon definition of the radicalization process or terrorism exists in international law [23].

The field of terrorism does develop risk profiles and has, for example, found that terrorists have very different profiles [24]. Both Maras and other authors have reasoned that somethings seems to happen in a subject’s life in the same time window as the radicalization process starts [24,25]. In the perspective of the TMT, fear of death can be assumed to be “this thing” that happens. Already more than a decade ago, MI5 suggested that terrorists are not loners and it is reported that rapidly escalating radicalization produces a higher risk that an individual actually executes a terrorist act [26,27]. The MI5 observation is a very interesting one because it is suggested that subjects who do not have high levels of the personality trait openness may have this trait further reduced when they are inflicted with EA [28]. The author suggests that comparable change related to other personality traits may be part of the change process subjects undergo when they are influenced by EA. The research from the last decade had documented that personality traits may not be as stable as previously assumed. Others have repeatedly argued that change in sexual matters is a potential risk factor associated with radicalization processes, and we know that EA is related to same type of change [29]. The existing knowledge about terrorism and other criminal profiles represent the definition of CPs in the present project. The important factor in ontology development is to identify overlap between knowledge about EA and knowledge about CPs. Radicalization is defined as the process were individuals or groups undergo a transformation process. During this process an increased support for violence and coercion to promote extreme views and beliefs is developed [24] (p. 4, p. 7); [30] (pp. 2–3). Detection of extremism is important even if the probability of terrorism is low, as the consequences when it does happen are very high. Radicalization can also be considered as a condition that reduces quality of life for those being radicalized and their relatives. Today, there is a continuous work in progress to identify risk indicators and treatment interventions to identify and cure radicalization [31,32]. Today, radicalization is not a diagnosis, and we do not really know the medical definition of radicalization, which makes both detection and treatment challenging. The goal of the present work was to combine findings related to risk from each of the four IBBB’s for EA with well-known CPs in an ontological framework. See Figure 1.

### 1.6. Literature Search

A literature search with a focus on the IBBBs and radicalization was undertaken in order to find out whether there are existing ontologies for the condition of EA and/or radicalization. One example of an ontology that combines neuroscience (world one or two) and behavior (world one) is the Neurobehavior Ontology (NBO) [33]. This ontology consists of two main components, an ontology of behavioral processes and an ontology of behavioral phenotypes, but does not include the condition of EA. Ontologies related to neuroscience and behavior but that do not include radicalization already exist. Furthermore, descriptions and ontologies describe how it is possible to move from brain maps to cognitive ontologies [34,35]. Cognition can be described as mental processes and can relate to Popper’s world two. Ontologies for brain maps already exist to a large degree, but no separate maps were found for EA and radicalization [36]. Many medical ontologies covering large or smaller areas of medicine already exist but do not cover EA and radicalization (see, e.g., [37]).

The possibility of using individual characteristics, especially personality, in ontologies has been examined before, as in the modeling work of developing personas [38,39]. For example, within user experience (UX) design, the persona characterizes a user within a segment. A persona is the development of a fake human (user) representing certain individual characteristics. However, a persona does not provide comparable levels of precision as does a personality test. For example, ontologies that use classical terminology relating to the big five personality traits were difficult to find. One ontology used personality related to fingerprints, but the information about the relationship between fingerprints and personality was scarce [40]. Ontological frameworks related to detecting extremists (field of terrorism) were found. However, all cover small parts of the field, and none cover all of the IBBBs and/or include EA [41,42,43,44,45]. In other words, ontologies that cover radicalization or extremism lack more traditional profiles in medicine and/or psychology. Others have investigated the existing ontologies related to human behavior recognition [46]. The article about behavior recognition contains a request for more accurate representations of everyday human tasks and language and better modeling of social interactions and environmental descriptions.

A free systematic search on the web was undertaken using the search engine Oria, delivered by the Norwegian University of Science and Technology (NTNU). The search words were “ontology and radicalization” or “ontology and existential anxiety” (replaced systematically by the words “mortality salience”, “fear of death”, and “death anxiety”, which are uses synonymously with EA) and “radicalization”. The searches were delimited by ontology, peer assessment, and English language. In total, the searches yielded 212 articles. However, only one article was relevant for the present subject, and this article was already previously presented (see Appendix A). The other articles found in the search were mainly about philosophy (including ethics and morals) and national and international security at the policy level.

The lack of articles presenting concrete modeling that can be used in practical work was striking. The literature search has detected no ontology of existential anxiety and radicalization. However, one ontology within psychology, “psychology ontology”, had included the class of death anxiety (but not specially related to TMT), which included the class mappings SNOMED (Standards for Health Terms in Patient Journals), two maps related to nursing practice, and two maps related to disorders [47]. Examples of classes that did include class mappings in the psychology ontology are behavior, personality, self-esteem, neurology, neuroimaging, neuroscience, neurocognition, neurochemistry, neurobiology, neuroanatomy, and neuropsychology. Moreover, this ontology also includes the classes of dopamine (1), Gamma-Aminobutyric acid (2), and serotonin (3), which are the three big “neurotransmitters” (“the three big chemistry” refers to three important substances involved in brain communication ) frequently related to aggression and criminal behavior [48]. The psychology ontology also includes the class of religious fundamentalism, but this class was empty (no class mappings). However, even if the psychology ontology lacks any class mappings for religious fundamentalism or radicalization, the ontology best matches the present work. Generally, it can be said that it seems that an ontology that consists of the four IBBBs related to EA and radicalization may not exist. The missing link is mappings of risk related to EA and how the IBBBs are related to radicalization.

This result suggests that already existing ontologies cannot provide information about the IBBBs for EA or how the IBBBs are linked to radicalization and criminal profiles. Examples of relevant databases related to radicalization processes or brain processing are the BrainMap project, the Global Terrorism Database (GTD), and libraries [49,50,51,52]. The GTD contains information about more than 190,000 cases. (For an overview of partly relevant sources found in the review process, see Appendix A).

### 1.7. Hypothesis and Idea of Present Work

This work hypothesizes that existing scientific knowledge about the IBBBs related to EA and CPs can be combined into a consistent, coherent system that can identify subjects based on certain patterns. Since no universally agreed-upon definition of the radicalization process or terrorism exists in international law, the present work rest on a condition described as one or more possible factors related to radicalization in the field of terrorism [23,53,54]. However, we have no documentation on whether this condition is related to violent terrorism. Therefore, the present subject concerns signs of being in the condition of EA given that other observations can confirm an overlap with EA and well-known existing indicators referred to as CPs. The idea is that by combining EA and CPs, it may be possible to identify early signs of radicalization. For example, a subject in the condition of EA may not represent any risk unless the subject can, in addition, be related to existing CP(s), which is assumed to be related to criminal offenses. Literature and data related to EA used in the present work are obtained under the “umbrella” of TMT, which is one of several types of existential anxieties. Furthermore, the ontology presents some frequently used and/or debated CPs.

Scientific knowledge is used to create an ontology and then test it using SPARQL Protocol and Resource Description Framework (RDF) Query Language. Results from queries together with some machine learning analysis is used to develop a sample-dependent profile of risk of early signs of radicalization/EA. The hypotheses are as following:Knowledge about the IBBBs and the CPs can be combined into a consistent, coherent system that can identify subjects based on certain patterns.SPARQL queries identify subjects in the condition of EA by separating those individuals from a control group.Results from SPARQL queries can be used to visualize a gap between early detection of radicalization and the probability of terrorism.The ontology can be reasoned to have a philosophical existence in the real world.

This process will be shown by using the software Protégé, delivered by the technical field at Stanford University, and Waikato Environment for Knowledge Analysis (Weka), developed at the University of Waikato, New Zealand [55,56,57]. The new versions of protégé are sophisticated and a sufficient tool to develop ontologies. The RDF query language is a semantic query language for databases that can create and manipulate data stored in RDF format to test whether the ontology identifies subjects based on some test data. SPARQL is a semantic query language for databases that is able to retrieve and manipulate data stored in RDF format [58].

## 2. Materials and Methods

This work uses research findings to build an ontology; it does not gather data but uses already existing data to test the ontology.

### 2.1. Sample

Data used in testing the ontology was drawn from eye tracking data of hundred and four participants, where fifty-two were in an experimental group, and fifty-two were in a control group.

### 2.2. Data

The ontology is built on findings of EA “under the umbrella of” the TMT and knowledge about some relatively well-known CPs. Findings are found in published scientific journals. Data gathered from 104 (24 of these of highest relevance for fixations and latency) participants were used to test the ontology. The stimuli material used in the experiment that gathered the data was of fetish or sensual quality (sensual/sexy man or a woman, a woman with a wipe, woman with a cross, sexy firemen, etc.) or neutral (a cup, a plate, rubber bands, etc.).

#### 2.2.1. Model Choice and Organization of Data on Protégé

This work is based on five selected models, where some variables relate to documented findings in academic studies and others are based on the author’s own research. Data for variables representing data from academic studies are simulated. Nevertheless, most of the data used to test the models are real data. For example, for the simulated values, each individual was given a true bool for radical/extreme values and beliefs. This fact applied to both the experimental group and the control group. By assigning all individuals at least one value related to CP(s), individuals from the control group are detected by the SPARQL query. It was assumed that the inclusion of a simulated constant for all would increase the difficulty of detection. The reason is that individuals can be radical and extreme without matching other indicators for being in a radicalization process. To qualify to be in a radicalization process, as many indicators as possible should match requirements, and here a threshold was set. The rationale behind the choice of four object properties is described in Section 3.3.

The reason behind choosing the models is complex and in large degree based on thorough investigation of eye tracking data for this condition. The literature contains, for example, a huge amount of documentation that self-esteem is related to different types of vulnerability depending on whether it is high or low. Unpublished data suggest that extraversion in addition to openness (mentioned in the introduction) may have a different risk profile, at least in subjects with high self-esteem. Some findings are part of a project that has been submitted for review at a journal some time ago. The brain path called focal processing can for simplicity be defined as the sexual path and many studies document that EA contributes to change with respect to sexual motivation. Appendix A describe the amount of evidence for each variable included in the models more exhaustively (also see Section 4.4).

#### 2.2.2. Test Sample(s)

Data used in testing the ontology were drawn from two samples of eye tracking (type described in the Section 2.3) data of 104 participants, where 52 were in an experimental group, and 52 were in a control group. What is special about this sample of data is that 80 of the 104 subjects had relatively high self-esteem. The sample consist of mainly young students (below 30) study computing, interaction design, psychology, nursing, or radiology. The personality traits for these 80 subjects were similar between groups (for an overview see Appendix A). A visual overview of the test procedure with respect to the data gathering process can be found in Appendix A. There is less information available for the sample of 24 subject, which had a somewhat higher average age of 28 years. In this sample the control group and experiment group scored similarly with respect to positive and negative affect. This sample consist of subjects with the same activity profile as the 80-subject sample. This suggest that the total sample only describes young people with academic potential and without any neurological disease that affects eyes. Moreover, the load is highest for subjects reporting moderate to high self-esteem. More studies have previously been published on individuals who have low self-esteem than on individuals who have high self-esteem. High self-esteem-subjects respond differently to priming with EA than low self-esteem subjects [59,60].

### 2.3. Instruments

The Protégé free, open-source platform for the development of knowledge-based applications was used to construct the ontology [56]. The OWL open source reasoner Hermit was used to validate the consistency and coherence of the ontology [61]. Quires were executed and developed using SPARQL (an acronym for SPARQL Protocol and RDF Query Language) [62,63].

The classification algorithm J48 in the datamining software WEKA was used to take random samples to examine how many variables are needed in a model [55].

The two samples of eye tracking data were gathered by using the following eye trackers:24 sample: Gaze data were recorded using an SMI iView XTM Hi-Speed 1000 (SensoMotoric Instruments, Teltow, Germany) eye tracker with sampling binocular gaze data at 500 Hz. The experiment was performed, and the initial data analyses were performed using SMI Experimental Suite 3.5. The data were further analyzed using IBM SPSS Statistics 23 (IBM, Armonk, NY, USA). For this sample, IBM SPSS Statistics 23 was used to aggregate the data.80 sample: Gaze data were recorded using an SMI (iView X™) Red 250 system (SensoMotoric Instruments, Teltow, Germany) eye tracker with sampling binocular gaze data at 250 Hz. The experiment was performed, and the initial data analyses were performed using SMI Experimental Suite 3.6. The data were further analyzed using IBM SPSS Statistics 25 (aggregation) and 26 (analysis) (IBM, Armonk, NY, USA). For this sample IBM SPSS Statistics 25 was used to aggregate the data.

### 2.4. Study Registration

These works were done as part of a project that started in 2014. The Regional Ethics Committee considered this study as non-clinical. Renewed contact with the Norwegian Center for Research Data (NSD) during spring 2020 confirmed at my request that the correct procedure had been followed. However, since data can be considered biological data, only some individuals (similar to real individuals) are made public (in ontology) in order to protect participants’ confidentiality [64].

## 3. Results

This section first presents the ontological framework. Second, it is “tested” whether the ontology can be used to detect individuals in the condition of EA by using SPARQL queries. The results from queries are further compared against an estimated risk of worldwide terrorism. The last part of this section presents an overview of the metric of the ontology. This section is divided by subheadings. The ontology, together with some example of individuals, can be found at DOI: 10.6084/m9.figshare.14195510.

### 3.1. Top Hierarchy of Ontology

The top hierarchy of the ontology is shown in Figure 2. The top hierarchy represents the main classes and the subclasses related to Individual characteristics, behavior, brain, biochemistry, and crime profiles (IBBB-CPs). In addition to the main IBBB-CPs, the otology has two other top hierarchy classes: group and environment. Groups describe arrays to which an individual can belong. For example, as shown in present ontology, for students or subjects primed with fear of death, the ontology is defined as CG (control group) and EG (representing EA). Environment refers to the milieu individuals operate in or settings where data is gathered or observed. These are environments consisting of all types of surroundings (e.g., war or peace). The focus in the present work is the five main classes, and the support classes will not be presented in detail because these classes must be adjusted according to the type of data for which the ontology is used. An example of how the group and environment classes are used can be found in the developed ontology.

### 3.2. The Five Main Classes IBBB-CP

In this section, the five main classes and their subclasses will be presented.

#### 3.2.1. Biochemistry

In present ontology, the biochemistry class is small and embraces only gamma-aminobutyric acid (GABA), an amino acid that functions as a neurotransmitter of the central nervous system and that inhibits excitatory responses. GABA is important for communication between brain cells, mood and relaxation. Serotonin is a neurotransmitter that is involved in, for example, sleep, depression, memory, and other neurological processes, and dopamine is a catecholamine neurotransmitter in the central nervous system and brain that helps to regulate movement and emotion. Brain biochemistry is closely related to genetics as, for example, investigated with respect to risk for violence, as well as with the aim of finding medicines that can be used in medical treatment (see, e.g., [65]). To the best of the author’s knowledge, only serotonin has, thus far, been related to EA [22]. An overview of the subclasses in the class biochemistry is shown in Figure 3.

#### 3.2.2. Behavior

The behavior class presents nine subclasses, which are types of behavior that are related to EA. These behaviors were chosen because they are well-documented findings or findings relevant to safety or security. For example, research findings related to sexual change and EA are found in many published articles [17,18,54,59,60]. Smaller pupil diameter is in the literature related to extremism [66]. The research findings related to familiarity and eye behavior is documented by the present author [54,59,60]. Acting out is a constructed behavior, as it is important concerning the execution of behavior acts due to radicalization. Via object property range and overlap, this class is related to the two main brain processing paths, as presented in the next subsection.

Avoidance and inhibition are in the literature related to EA [67,68,69]. The preference for familiarity is related to EA, for example, the preference for ingroups compared to outgroups [59,60,70]. Familiarity is in the ontology related to the CP via object property range to a blank stare [71]. The literature suggests that EA may escalate psychiatric symptoms, which affects behavior, brain processing, and biochemistry [72]. However, the literature does not document exactly how this escalation could hypothetically affect radicalization processes. However, we know that certain psychiatric diseases may be related to increased risk of violence [73,74,75,76,77]. The literature also documents that EA is related to objectification [19]. The literature and the research findings related to EA and behavior are comprehensive and cannot be exhaustively presented here. For an overview, the article by Brian L. Burke [17] is recommended. However, research findings about EA and physical behavioral indicators that can be used in computing, for example, are scarce. An overview of the subclasses in the class behavior is shown in Figure 4.

#### 3.2.3. Brain

The subclass brain refers to brain processing or parts of the brain relevant for EA and radical values and beliefs. Insula and body awareness are two subclasses that overlap in the ontology. This overlap is related to research the shows that EA is related to reduced body awareness and reduced activation in insula [78,79,80,81,82,83]. The areas in the brain called the ventrolateral prefrontal cortex (VLPC) and orbitofrontal cortex (OCF) have been found to have enhanced activation to death-related stimuli [82] (also see Reference [84]). Self-esteem is also found to moderate the relationship between the amygdala and the VLPC region in response to mortality threats [21]. The most interesting aspect of these two findings is that OCF and VLPC changes seem to relate to low self-esteem, while increased amygdala-VLPC connectivity relates to high self-esteem (also see Reference [54]). This information is included in the object properties as a range. The subclasses of the class brain processing are shown in Figure 5.

Studies have found that fear of death influences risk perception, for example, in a virtual environment (VR) related to the distinction of risk perception of self or others [85,86]. Risk perception is described together with ambient and focal processing because these visual processing paths are frequently investigated related to safety [87]. Ambient and focal processing seems to change in the condition of EA, yet the base of knowledge is small [54,59,60]. Ambient and focal processing is important because these paths are related to pre-motoric processing and sexuality [88,89]. These paths are, therefore, strongly related to the subclass of behavior.

#### 3.2.4. Individual Characteristics

This subclass refers to attributes that can be related to individuals. Agreeableness, conscientiousness, extraversion, neuroticism, and openness to experience are all personality traits. These five personality traits represent the traits inherent in the most often used psychological trait theory called the big five or the five-factor model (FFM). Robert McCrae and Paul Costa developed this theory based on research published by others, such as Gordon Allport, Raymond Cattel, and Hans Eysenck [90]. Personality traits are easy to detect and can, for example, be detected from online activities, such as Facebook use or eye-tracking [91,92]. This class also includes the subclasses of age, gender, and locus of control (LoC). For example, EA has been found to increased risk-taking in subjects with external LoC [86]. Individuals with internal LoC had reduced risk-taking when under the influence of EA. LoC is a psychological concept that refers to how strongly people believe they control the situations wherein they find themselves and their lives, or whether the environment directs their control. The concept of self-esteem as a description of our subjective evaluation of our own worth has existed for a very long time. The field of psychology, a relatively young academic discipline, has further developed the concept and designed clinical practice tests. Self-esteem has been strongly related to the vulnerability for EA according to the TMT, but this position is debated [18]. Gender is included as a subclass, but EA is not strongly related to gender differences [17]. Age is a subclass that overlaps with age as a risk factor for extreme CPs’ extreme behavior. An overview of individual characteristics can be found in Figure 6.

#### 3.2.5. Crime Profiles

The class crime profile has no meaning in this ontology unless certain subclasses can presently or in the future be related to the other classes of IBBB. In summary, the classes of IBBB and CP yield IBBB-CP, which should describe the possible relationship between known criminal profiles and EA or declare that no such relationship exists. The relationships between EA and CP in this model are only partly confirmed by documentation in the literature. Today, we do not know whether behavior in the condition of EA resembles lying or whether EA is related to psychosis. These attributes are still under investigation (see, e.g., this poster [93]. Whether this condition may resemble a blank stare is under investigation by the present author, and preliminary findings suggest that simulation of eye behavior shows that EA may resemble a blank stare [60]. EA is found to be related to inhibition, which can predict overcontrol [68,94,95]. Whether CPs are related to criminal offenses is debated, for example, a blank stare/staring gaze is debated (see, e.g., [71,96,97]. Extreme behavior, which is related to extremism, will be described separately in the next section. An overview of the subclasses of the CP-class can be found in Figure 7.

#### 3.2.6. Extreme Behavior

Many attributes related to extreme behavior can be found, for example, on the website of the National Consortium for the Study of Terrorism and Responses to Terrorism (START) [27]. For example, the relationship between sexuality and radicalization is mentioned by Kruglanski and colleagues and can be related to the focal brain processing path [29,88]. The present author has found that for subjects primed with EA, the focal and the ambient (pre-motoric) paths may change compared to controls [54,59,60]. Many indicators found for extremism are also considered indicators of illegal activity: lack of integration, having a criminal record, history of abuse, affiliation to criminal networks, psychiatric disease, drop in social standing, and age. Therefore, what is to a large degree new regarding radicalization is change in sexual attitudes or behavior, smaller pupil diameter, change in performance, and the possible rapid escalation of a radicalization process. The subclasses of extreme behavior are shown in Figure 8.

#### 3.2.7. The Relationships between the IBBB and the CPs

This section will be devoted to presenting examples of relationships in ontology. First is the overlap between age in the individual characteristic class (I) and the age related to extreme behavior in the CP class. Extremism is related to a relatively young age. This point is shown in Figure 9.

Figure 10 shows the relationship between the three main biochemistry processes in CP and the biochemistry related to EA in one of the Bs.

Figure 11 describes the relationship between the change in sexual behavior associated with EA and the change in the same variable related to radicalization. Concerning this relationship, it is worth mentioning that EA is related to different development depending on individual characteristics, such as self-esteem and neuroticism/anxiousness [18]. The research suggests that EA most frequently increases the search for sex and partnership, while the overlap with CP is based on avoidance or reduced activation of the sexual path.

In Figure 12, we see an example of possible connections related to acting out (B) and an extreme behavior (CP).

Present ontology presents a framework where the sexual path decreases, and the pre-motoric path increase based on research done by the present author [60]. This pattern is assumed to overlap with suggestions made in the field of terrorism [29]. This information is shown in Figure 13.

### 3.3. SPARQL Queries

The present results refer to the queries described in Appendix A. The results are a sample-based estimate based on results from the SPARQL queries. What is the probability that subjects from a relatively random population should be seriously radicalized to such a degree that they “need help”? The ontology detects subjects according to a “dualistic design”, as either true (1) or false (0), which can be defined as Bernoulli or binomial distributed probability. In the introduction the probability that an individual in the world commits terrorist act in the world was given as 1.1 $\times \;10^{-6}$, and the average probability from queries in the five models as presented in Table 2 is 0.05, or 0.02 if considering the whole test sample.

Random generating of a Bernoulli distributed sample of 100,000 based on the respective probabilities was used to develop Pareto charts for the five models and the risk of terrorism. A Pareto plot shown in Figure 14, visualizing how use of extreme-event probability distribution may contribute to dwindling down the field of prevention. The figure shows that the probability of detecting a subject who is a terrorist through the ontology is extremely low. The distribution based the estimate for terrorism does not present a single case in a sample of 100,000.

Figure 15 shows a binomial distribution of the model probabilities and a cumulative distribution function (CDF) plot based on model probability and the estimated risk of terrorism ( 1.1 $\times \;10^{-6}$). This figure visualizes the exponential reduction in probability of detecting early signs of radicalization by using a probability measure of extremely rare event0. The figure visualizes the probability of NOT identifying radicalized subjects. The CDF refers to the probability that X takes a value less than or equal to α, and the probability of the input distribution is the inverse of positive probability distribution (e.g., 1 − 1.1 $\times \;10^{-6}$ = 0.99 or 1 − 0.05 = 0.95).

Each of the sample models has at least four variables included, and this design was made after predictive testing using machine learning. Previous analysis has been done on this material by using other machine learning algorithms, but the present analysis uses the Weka algorithm J48 based on models that are known to be significant. The results from these tests are shown in Table 3.

Finally, the Bernoulli distributed files were placed together into machine learning to shortly test whether the random numbers based on the estimated probabilities from queries could preach at all. The accuracy started at 85.3% without doing any optimizing classifications or data mining (e.g., Sequential minimal optimization (SMO) or simple logistic). Machine learning classification is outside the scope of this project, but results from queries can be used in machine learning analysis. In summary, even by including simulated values about the relationship to criminal profiles, together with real measures for EA, detection of a compound/set of indicators will be relatively rare. This is even if the condition of EA is a condition humans frequently experience.

### 3.4. Ontology Metrics

The ontology metrics in Table 4 shows that the ontology at moment consist of 150–173 classes (depend on ontology version), 84–103 data properties, and 60–66 object properties. The numbers of data properties are high because some values were intended to be used at different levels, for example, to give space for different types of metrics. The biggest challenge in creating an ontology is to limit the number of classes and properties and this is a slimmed down version compared to starting point. Values for individuals are manipulated with respect to gender, age, exact values and simulated values in the public sample. This is done to ensure the participants’ confidentiality. Test data originally had hundred and four individuals of, which 80 had highest relevance with somewhat different values.

## 4. Discussion

This work shows that it is possible to develop an ontology that involves information from various sources from different disciplines. However, one could ask whether it exists.

### 4.1. A Three World Framework

An ontology is a world three framework produced by humans and will not exist in the world of monists and dualists. The existence of the present ontology demands a world three perspective that acknowledges the existence of human products and abstractions. On the other hand, it could be argued that the queries made by using the ontology may exist depending on object properties and data types, but this will be a rare occurrence and can be debated, because queries frequently ask whether something is true or not. The ontology detects individuals with more extreme values/beliefs based on the published documentation (world three) related to EA. The ontology may detect an individual in the early beginning of a radicalization process such that this individual can be offered help to change their values/beliefs (world two). By offering such help, the ontology (world three) can contribute to changing the individual (world one) and, in this way, have a causal effect on the world (world three objects). More concretely, the relationship between the IBBBs and the CPs (world three objects) in the ontology contributes to establishing a hypothetical causal effect on the world. The ontology and its queries qualify as having a world three existence according to Popper’s description [10].

### 4.2. Detection a Rare Occurrence

The results from queries presented in Section 3.3 show that the probability of detecting a subject in the condition of EA is rare, even if data used in the queries for some variables show a statistically significant difference between EA-group and control (50% are influenced by EA). The reason for this is that acceptable predictions about psychological conditions demand a compound of predictors. Here, we operate with a compound of four predictors, which is at the low end.

The IBBBs contain subclasses of “attributes” that the scientific literature has documented to change due to EA or radicalization processes. Regarding predictions and the fields of medicine and psychology, it is not acceptable to define a mental condition according to a simple proof of one dimension of a condition. This point is, for example, shown very clearly in the medical and psychological framework for making diagnoses. There are good reasons to suggest that the same should apply for risk predictions. Therefore, this section will mainly exemplify queries with more than four object properties. Moreover, the present work can only use real data from individual characteristics (all personality traits and self-esteem) and eye-tracking measures. Other data sets with measures for EA were not found after a search. The queries are based on real data as far as possible. The machine learning testing showed that exclusion of individual traits’ variables, as well as a small number of variables, reduced the accuracy considerably. This suggests, firstly, that some individual characteristics should be included in the ontology query model. Secondly, it should include as many contributing variables as possible, and minimum four, which include at least two trait variables. Within health research, it is not professionally acceptable to draw conclusions based on one or very few values. We can see why test batteries developed to make diagnosis are voluminous. That EA contributes to more extreme values and beliefs is highly documented in the literature [98]. Thus, given that we know that one individual is in the condition of EA, it is reasonable to assume that this subject has more extreme values and beliefs than those not in the condition of EA. However, in real life outside of an experimental setting, we do not know whether a subject is in the condition of EA other than through physiological and psychological observable indicators. The results from the query confirms that even if 50% of the subjects are known to be inflicted with EA, the probability of detecting “risk subjects” is low. The prevalence of detection will probably be even lower in a real-world setting. The present ontology is built on knowledge about EA and not on the number of extremist acts. The fundamental idea is that it is preferable to detect radicalization at an early stage before any violent tendencies so as to offer help and support. Nevertheless, queries can be a useful exercise, as shown in Figure 14 and Figure 15. Queries may reduce the “area/region of interest” and, at the same time, contribute to the detection of early signs of radicalization before any escalation. Importantly, these figures also shows that searching for radicalized subjects based on terrorist probability will not contribute to the detection of early signs of radicalization. In searching for extremely rare events, the smaller signs will not reach a level of significance that makes these signals of importance. The results suggest that looking for early signs of radicalization may be profitable with respect to preventive interventions. Hypothetically, it can be advantageous to look for small signs because a bias for extreme expressions can contribute to the bypassing of small signals. Cognitive bias is a natural part of risk management work, but the work itself is investigated as a risk factor less frequently. Working with “the extreme” does affect human beings. Figure 15 in Section 3.3 exemplifies this further by showing the different profiles of a binomial distribution of the models, as well as a CDF plot of the exponential development of not detecting signs of radicalization or terrorists.

In summary, detection of a “radicalized” individual using queries as described in Section 3.3 above is rare, but it is more frequent than terrorism. This result underscores how difficult it is to detect people based on their psychological state. Furthermore, the results indicate that to exclusively search for extreme events may bypass small signals such that potential radicalization may not be detected.

### 4.3. The ’Problem’ of Distribution

Estimating rare events are difficult and it has recently been proposed that extremely rare events should be estimated using a Bayesian-Bernoulli-Weibull mixture [99]. The thinking is that if subjects do not share traits that qualify for inclusion at all then they should not be included. This argument can be debated depending on where the cut off is set. Results from the present analysis suggest that one may not operate according to the probability of extremely rare events if the aim is to prevent them. The ontology may function as a gatekeeper such that subjects who do not have one or more of the CP-risk factors are included in the risk distribution. This suggests that the ontology can be used as an initial probability testing for early signs of radicalization and not as a probability that someone is a terrorist. Reduction of a risk population is important to finetune investigation and medical or psychological interventions. The probability from ontology queries is low, but not too low to detect early signs. This leave the distribution ‘problem’ unanswered because the probability is not low enough to qualify as an extremely rare event and probably not high enough to qualify as being part of normal everyday life. The ontology detects very few instances because in the five models many of the same individuals were detected. Based on the analysis presented in Section 3.3, it is not totally clear where this level of risk belongs in terms of distribution. Since this is, to the author’s knowledge, the first ontology made based on the present design, time may show how the distribution of EA and the distributed relationship between EA and radicalization may develop.

### 4.4. Generalizability

The present ontology is built based on research publications that document how EA contributes to change related to the IBBBs and relates this information to CPs. For some classes, there is a large amount of change due to EA, such as radical values/beliefs, self-esteem, inhibition, sexual or mating behavior, and avoidance of emotional content. However, concerning biometric measures, for example, pupil diameter, eye tracking, and the other traditional biometric measures, the documentation is scarce or non-existent. The exception is cognition and brain processing, where the amount of documentation has improved in recent years. This situation suggests that the ontological framework is based on varying degrees of evidence, and that there is much we do not yet know. For example, we know from many studies that EA is related to change in sexual or mating behavior, but we still know little about how this is related to brain processing paths. Nevertheless, when this is said present work have support in literature and data. Evaluation of the generalizability of each variable included in model I–V is described in the Table 5.

Because of the varying degrees of documented evidence related to EA, the ontology must be considered an ontology that can be further developed and hopefully based on more scientific evidence. Moreover, the Ontology Framework to Facilitate Early Detection of ’Radicalization’ (OFEDR) ontology is not tested on samples consisting of subjects known to be radicalized. It is necessary to test the ontology on radicalized samples before using it in a professional setting. However, this will probably be a relatively simple matter for researchers who have access to such samples. Moreover, brain coordinates are not tested or fully implemented in the present ontology because the present work has no access to data samples of brain coordinates related to EA. Furthermore, suppose coordinates are retrieved from different stereotactic spaces. In that case, it is necessary to transform coordinates to one stereotactic space, which can be executed by using software, such as Neurosynth and Nimare [34,100,101,102].

In summary, the ontology detects test subjects according to the level of knowledge related to security and safety in EA combined with some existing CPs.

### 4.5. Prohibition, Screening and NOT Decision

Profiling can only be used within the legal regulations of international law, the General Data Protection Regulation (GDPR), and national law. According to the GDPR, profiling people without their consent is not allowed [103]. However, today, a large amount of open-source data blurs the boundaries of private data and even the intelligence agencies (counterterrorism, LEA) to a large degree use data from open sources [104,105,106]. However, profiling data about individuals gathered through open sources cannot, according to the GDPR, be used in profiling without informed consent. This context suggests that the ontology can be used mainly on research data or data other than informed consent (e.g., special legislation).

This ontology is based on scientific evidence, which is partly still weak and should, therefore, not be used as the only source for making decisions. Further, testing of the ontology is necessary before its use in professional settings. The ontology intends to serve as a screening tool prior to a more thorough examination, and not as fundamental for making decisions. Within, for example, medicine and psychology, screening instruments should never be used as the only source of information for making decisions.

### 4.6. ’From Individuals to the Level of State’

The OFEDR ontology has a few top hierarchy classes, making the ontology easy to reuse and change. The ontology shows to a practical and detailed level how knowledge from different disciplines related to EA can be related to existing criminal profiles. The detailed and practical level divides this ontology from ontologies presented at the policy level and/or network level, including geographical profiles. To the best of the author’s knowledge, no ontology comparable to OFEDR exists. However, the effect of EA is documented in a considerable number of studies (500 reported in 2014, and many thereafter). Ontologies can be used as a tool to investigate and prevent crime [107]. However, the literature search undertaken related to this work showed that radicalization, especially terrorism, is mainly investigated at a higher policy level [108]. Some articles contain concrete and practical object/object properties related to radical behavior, but they are still very few. The Manshaus case mentioned in the introduction shows that radicalization should concern everyone, since this condition can appear anywhere, even within the four walls of a house [1].

Stuart Croft [108] refers to an important innovative shift in the development of ontological security studies, which includes a change of focus from individuals to the level of the state. The literature search undertaken in this work may confirm a strong focus on state security, policy, and philosophy. Furthermore, as frequently found, a traditional individual focus, for example, in psychology, was completely absent. However, it seems that the field of terrorism is in the process of shifting its focus in the direction of placing more emphasis on physiological and psychological indicators [109]. The literature review indicates that the focus on radicalization/terrorism research has been more strongly directed at national and international policy level than on very specific risk assessments that can be measured at a lower or individual level.

One possible reason for the lack of ontological focus based on individual characteristics may be the bad reputation that the field of profiling gained due to a lack of reliability and validity. However, today’s profiling is very much improved compared to earlier times [110]. Profiling has also been related to political controversies considered threats to human rights to freedom. These viewpoints are understandable if profiling is considered as a final answer or are relied upon uncritically. We have examples from the diagnostic system within psychology showing that such systems cannot be uncritical. For example, it has only been a few years since homosexuality and deviant sexuality that does not harm or stress anyone was removed from the psychological diagnostic manual (DSM) [111,112].

The need for security will always compete with human rights. The present ontology cannot be used lawfully by unauthorized individuals to profile subjects without their consent. To a large degree, human rights are protected and regulated, and knowledge including systems to improve security is well regulated. Why do we need a framework that can only be used according to the limitations of the law?

EA is a condition that activates at an unconscious level. We may not know that the condition influences us. This fact suggests that we need knowledge and systems to detect early symptoms before they escalate. Another argument is that EA is found to worsen already existing psychiatric symptoms [72]. Moreover, today we do not have a uniform agreed-upon definition of radicalization or terrorism [23]. To a large degree, we still do not know what radicalization/extremism is. Therefore, all types of knowledge and systems related to these subjects are treated with cautiousness and curiosity. This lack of knowledge also shows a great need for more research that concretizes what radicalization is and how people become radicalized, as well as how such radicalization can be detected.

### 4.7. Assumptions

The OFEDR assumes that three world objects exist. Another assumption is that EA is related to political radicalization, as documented in studies published in academic journals. The risk of terrorism is estimated based on publicly available measures and does not represent an official metric for the risk of terrorism worldwide. The framework relies on research findings that may change in the future due to further development of the knowledge base through scientific research.

### 4.8. Limitations

It is a limitation that the present ontology is based on scientific knowledge with varying degrees of scientific evidence. More research is needed, especially related to the development of measures in the disciplines of biometrics and physiological behavior. Further, it is a limitation that the data used to test queries are not real-world data but eye-tracking data from an experimental setting. Moreover, personality and self-esteem measures are only “recorded” for 80 of the 104 subjects. The data are from subjects inflicted with EA and not from subjects involved in a known radicalization process. The data sample used in testing contains subjects reporting relatively high self-esteem. Moreover, some of the CPs are heavily debated [71]. For example, analyses (by threshold in ontology) showed that the object property a blank stare is not reliable for use as the sole measure. In general, the use of individual indicators should be avoided.

The knowledge used in the ontology is from different disciplines and does not cover all possible knowledge about each subject. The ontology requires more testing on different samples before use in professional settings and should be treated with caution. First and foremost, the ontology should preferably be used in research and other settings where informed consent is obtained without making any “diagnostic decisions”.

## 5. Conclusions

This work presents an ontology that uses information from various sources from different disciplines and combines the information to predict whether a given person is in a radicalization process. SPARQL queries show that the probability of detecting an individual in the condition of EA/radicalization is low. Given that the queries are done on a sample of experimental data where half are inflicted, and the other half is a control group, the probability of detection is surprisingly low. It may be reasonable to suggest that the detection probability would be even lower in a real-world setting. The results from the queries suggest that detection of early signs of radicalization may be possible but is a challenging task when combining indicators. Testing suggests that it is a prerequisite that emphasis be placed on a compound of objects/object properties and not only on one or a few attributes in predictions. The analysis shows that relying on strict measures for rare event probability based on the risk of terrorism reduces the possibility of detecting early signs of radicalization. This suggests that relying on strict measures for rare events may not contribute to early detection of risk subjects. Because those subjects will never appear in the sample due to a high threshold of inclusion. It can be concluded that SPARQL queries could identify subjects in the condition of EA. The probability from ontological modeling is low, but higher than the probability of a terrorist act, and this gap may represent a region of interest in the work of preventing escalation of radicalization processes. It is reasoned on a philosophical level that the ontology exists even if it is a level three object. The results from SPARQL queries suggest that the ontology may contribute to reducing a possibly existing gap between early signs of radicalization and estimation of extreme events, and thereby streamline preventive work.

## Figures and Tables

**Figure 1 jimaging-07-00060-f001:**
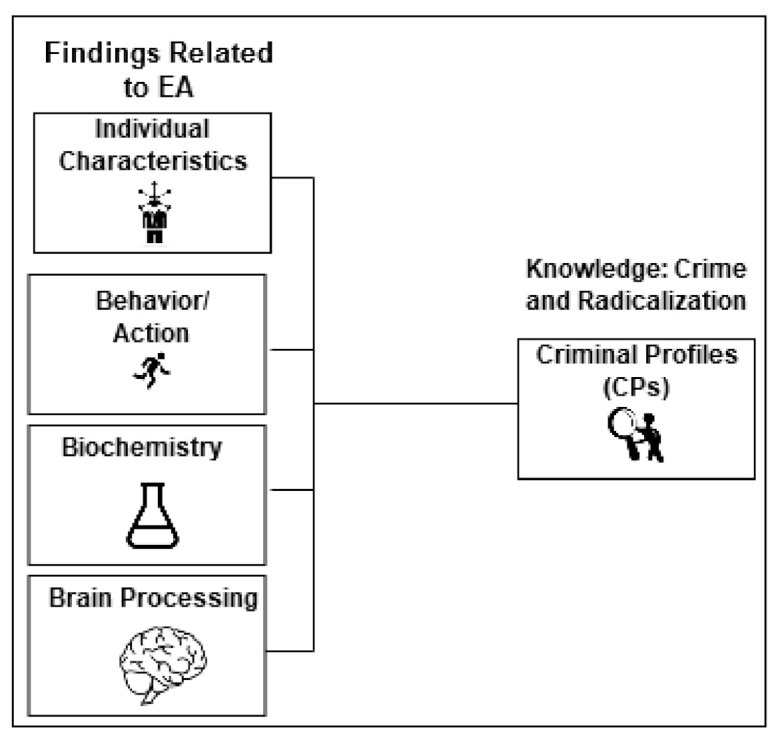
Ontology idea based on combining the four IBBB’s (individual characteristics (world level 2 or 3), behavior (level 1 or 3), biochemistry (level 1), brain processing (level 2)) and CPs (criminal profiles) (level 3).

**Figure 2 jimaging-07-00060-f002:**
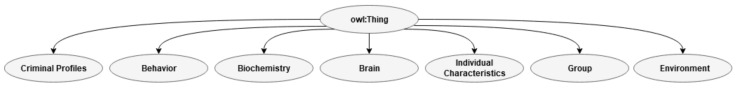
Top hierarchy of ontology. Behavior (B), biochemistry (B), brain (B), individual characteristics (I), and criminal profiles (CPs) represent the five IBBB-CP, while group and environment represent support classes.

**Figure 3 jimaging-07-00060-f003:**
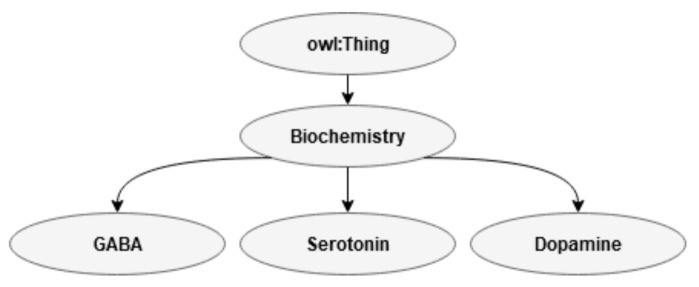
Class biochemistry.

**Figure 4 jimaging-07-00060-f004:**
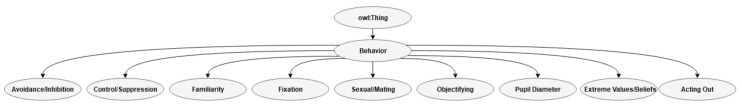
Class behavior.

**Figure 5 jimaging-07-00060-f005:**
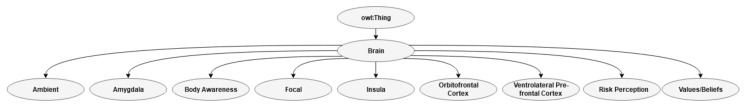
Class brain processing.

**Figure 6 jimaging-07-00060-f006:**
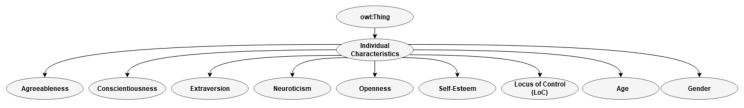
Class individual characteristics.

**Figure 7 jimaging-07-00060-f007:**
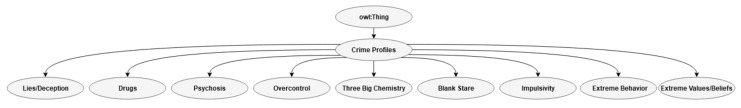
Class crime profiles.

**Figure 8 jimaging-07-00060-f008:**
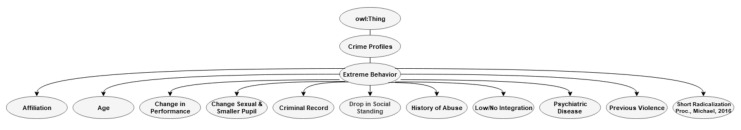
Subclass Extreme behavior.

**Figure 9 jimaging-07-00060-f009:**
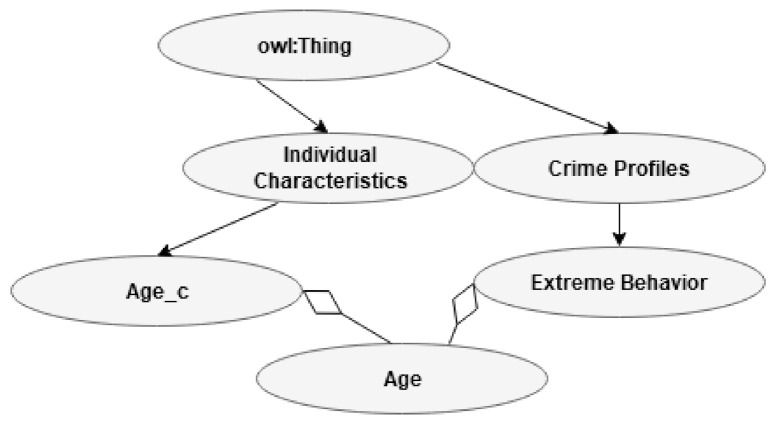
Relationship between age (I) and age (CP).

**Figure 10 jimaging-07-00060-f010:**
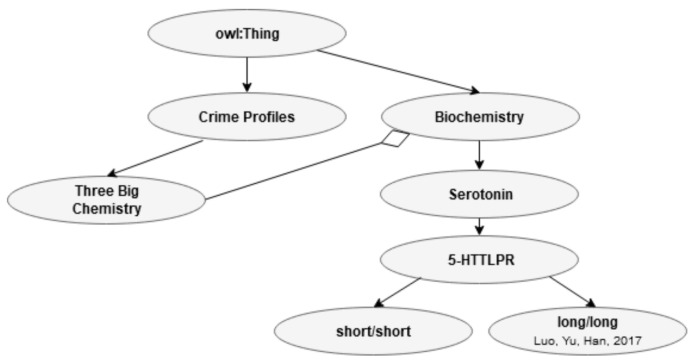
Relationship between biochemistry (B) and biochemistry (CP).

**Figure 11 jimaging-07-00060-f011:**
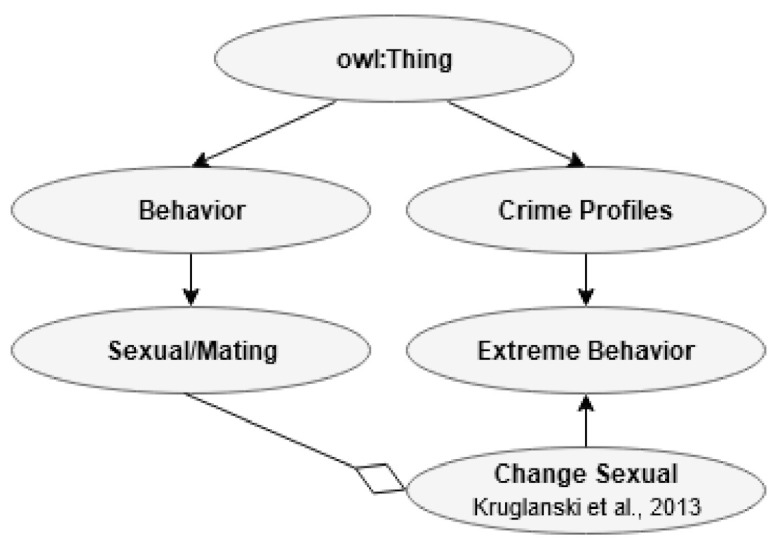
Relationship between sexual change related to EA (B) and the sexual change related to extreme behavior (CP).

**Figure 12 jimaging-07-00060-f012:**
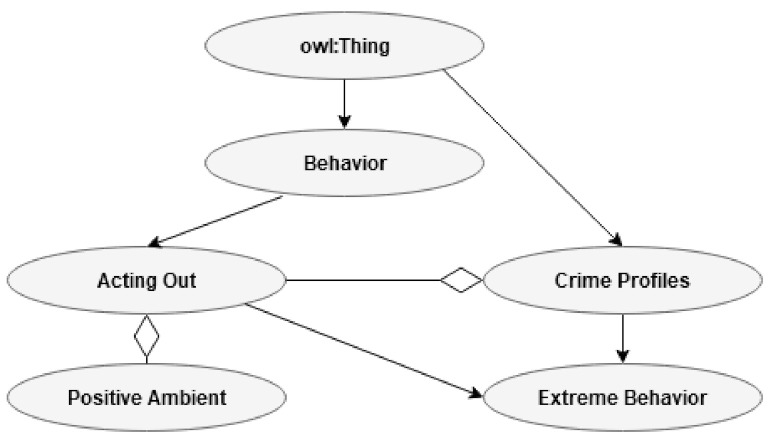
Relationship between acting out in one of the B’s and extreme behavior in CP.

**Figure 13 jimaging-07-00060-f013:**
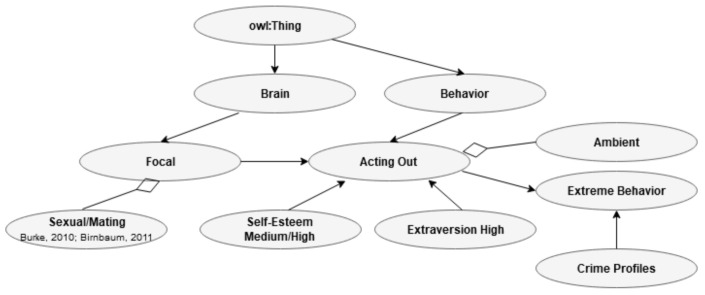
The relationships between acting out in one of the B’s, individual traits (I), and extreme behavior in CP.

**Figure 14 jimaging-07-00060-f014:**
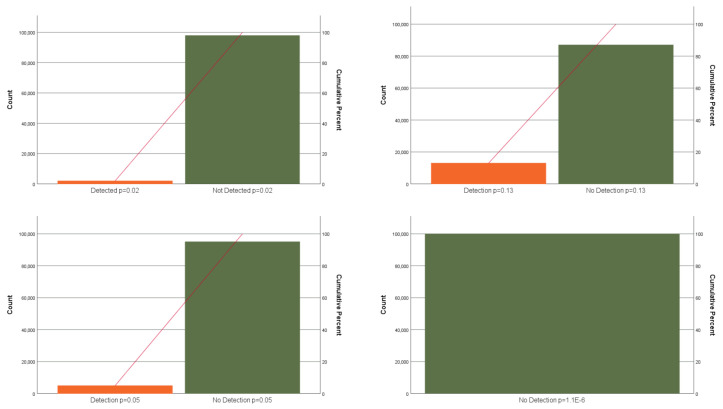
Pareto plots of Bernoulli samples of 100,000 based on the respective probabilities. Top left *p* = 0.02, top right *p* = 0.13, bottom left *p* = 0.05 (is also model average), and bottom right *p* = 1.1 $\times \;10^{-6}$.

**Figure 15 jimaging-07-00060-f015:**
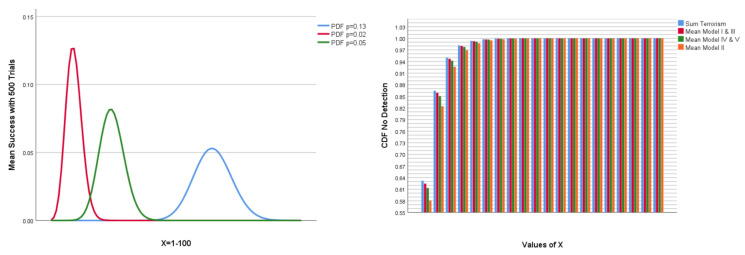
The figure on the left shows a binomial probability density function (PDF) for the p-values of the five models. Number of trials is 500. X starts at 1 and ends at 100. The figure on the right shows an exponential cumulative distribution function (CDF) plots no detection based on the respective probabilities: Model I–V (0.02–0.13) and terrorism (1.1 $\times \;10^{-6}$) (X = 1–10).

**Table 1 jimaging-07-00060-t001:** Popper’s three ontological worlds.

World	Defintion	Example Entities	World View
One	physical objects	non-living physical objects,	monism
		living things, optical fibers,	
		individuals, memory traces	
Two	psychological	feelings, pain, pleasure	dualism (world 1 and 2)
	objects	subjective experience,	
		suffering, conscious and	
	mental processes	subconscious experiences,	exist through world one
		thought processes, beliefs,	
		subjective states	
Three	products of	religious myths, songs, art	threefold realism
	human mind	theories, articles, airports	
	objects	information, language	logical relationships between
		thought content, qualitative	world three objects
		evaluations, true or false	
		(bool), e.g., belief, plans of	causal effect on world
		action, maps, computer	
		programs, plans	gives world two power to
			change world one
			is shareable, can be criticized

**Table 2 jimaging-07-00060-t002:** Results from SPARQL queries for model I–V. *, e.g., related to a blank stare ** change sexual and pre-motoric *** indirect through relationship between inhibition and overcontrol **** focal and ambient are two different variables (counted as 1 or 0). I = Individual characteristic, B = Behavior/Brain processing/Biochemistry and CP = Criminal Profiles. Probability column shows probability based on (a) numbers of subjects primed with EA (p) (b) all subjects (p) written as p/p.

Model	Variables	Type	Probability p/p
**I**	Pupil Diameter	B	0.02/0.01
	Fixation	B/CP*	
	Ambient and Focal ****	B/CP **	
	Radical Values/Belief	B/CP	
**II**	Pupil Diameter	B	0.13/0.06
	Ambient and Focal	B/CP **	
	Radical Values/Belief	B/CP	
**III**	Pupil Diameter	B	0.02/0.01
	Latency	B/CP *	
	Age	I/CP	
	Low Self Esteem	I	
**IV**	Pupil Diameter	B	0.05/0.01
	Low Self Esteem	I	
	Body Awareness	B	
	Serotonin	B/CP	
**V**	Pupil Diameter	B	0.05/0.01
	Inhibition	B/CP ***	
	High Extraversion	I	
	High Self-Esteem	I	

**Table 3 jimaging-07-00060-t003:** Testing numbers of model variables.

Numbers Personality Metrics	Numbers Eye Behavior Metrics	Numbers Total	Accuracy Level	Comments
6	1–5	7–11	>95%	Big five personality traits
				and self–esteem.
0	1–6	1–6	50–68	Lack of individual traits
			max 79	produce accuracy drop.
				Other models may produce
				higher accuracy but not
				very high.
3	2	5	⩾95%	Tuned variables.
2	2	4	84–90%	Tuned variables.

**Table 4 jimaging-07-00060-t004:** Ontology metrics. * Individuals represent persons with manipulated values. Their gender and exact numbers does does not match actual values obtained in the data acquisition process. No subject had the same numbers e.g., number 107 and 109 etc. does not exist in the “batch”.

Type	Metrics
Axiom	2084–2216
Logical axiom count	1673–1762
Declaration axioms count	331–381
Class count	150–173
Object property count	60–66
Data property count	84–103
Subclass of	144–165
Individuals *	34–36

**Table 5 jimaging-07-00060-t005:** Generalizability. Science refers to scientific literature.

Variables	Reasoning	Summary Generalizability	Test/Science
Pupil	Scope of evidence limited. Small	Can be generalized to some degree	Science & Test
Diameter	pupils are related to extremism [66].		
	Large pupils“=”high emotion.		
	Small pupils“=”low emotion.		
	Stimuli type should be considered.		
Radical	High evidence EA and terrorism.	Can be generalized	Science
Values/			
beliefs			
Ambient/	Low evidence direct. High evidence	Can be generalized to some degree.	Science & Test
Focal	indirect, e.g., change in sexual motiva-	Indirect findings must be considered.	
Paths	tion. Evidence in test data.		
Fixation	Does change in EA [59]. Depend on	Can be generalized to some degree.	Science & Test
	stimuli type, e.g., familiar or novel.	Cannot be used as a sole measure.	
Latency	Few evidence. Average latency (emot-	Can be generalized to some degree.	Science & Test
	ion) may be more important [59]		
	than onset of first saccade (impulsiv-.		
	ity) Stimuli dependent.		
Age	Age matters EA and terrorism.	Can be generalized	Science & Test
	Test show no difference because		
	all subjects are young (mean).		
Self-Esteem	Many studies. Test sample (n = 80)	Can be generalized	Science & Test
	had no subjects with low self-esteem.		
Serotonin	One study related to EA [22].	Can be generalized to some degree	Science
	Studies about other aspects as, e.g.,		
	burnout may support relationship.		
Inhibition	Relatively high evidence.	Can be generalized	Science
Body-	Good evidence. Findings related to	Can be generalized	Science
Awareness	insula (brain region) may support		
	The field of terrorism may also provide		
	evidence. Violence is less aware.		
High	Few evidence. Supported by work	Can be generalized only based on test data	Test
Extraverion	made by present author [60]	Young high self-esteem subjects	
	Compare findings about openness [28].		

## Data Availability

This study develop an ontology and does not gather data. Data is used to test the functionality of the ontology. Some data from eye tracking (is) may be made available at figshare https://figshare.com/s/b4b5b206efeb27f395e6 (doi:10.6084/m9.figshare.14195510).

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
