# Peer review of "An Ontological Framework to Facilitate Early Detection of ‘Radicalization’ (OFEDR)—A Three World Perspective"

_2313-433X, 2021, doi:10.3390/jimaging7030060_

Round 1
Reviewer 1 Report
-In 273 indicate what is GABA
-In the paper you should explain how they made the eye-tracking
-Between 394 and 395 In the queries can be found in Table A1 in Supplementary section. I think that is Table 2 supplementary section
-The relationship of equation 1 with the ontology and sparql queries is not understood
Could you explain more equation 1, table 1 and figure 14
-In 2.3 you should mention that each query is a model that is just observed in appendix 2. Query examples
-It does not place the OFEDR ontology or any of its parts as classes, subclasses and properties
What software did you use to build the ontology?
-You should specify more about the ontology metrics in the text of the article only mention a table in appendix 3
-In the conclusions section you mention that the literature review is a product of the results of the work but it is part of the literature search

Author Response
Dear Reviwer,
Thank you for your input to the manuscript. I will reply to each of your comments step by step.
English proofing is done on the entire manuscript.
-In 273 indicate what is GABA I have changed the creation of abbreviation to the section you refer to. GABA was previously was made to an abbreviation in the introduction. In addition, I have written more about GABA. These three chemicals are of great importance for our brain function. See also small changes made in introduction to explain the three big chemistry.
-In the paper you should explain how they made the eye-tracking I have made a broader description in the method section and added a visualization figure in the supplementary section (Figure A1). The projects with eye tracking and its design is that I use fetish stimuli and repetition of stimuli. Therefore kan different processing of novel and familiar stimuli be detected.
Between 394 and 395 In the queries can be found in Table A1 in Supplementary section. I think that is Table 2 supplementary section Yes, it was two errors with respect to numbering of this table. I have hopefully changed it correctly now.
-The relationship of equation 1 with the ontology and sparql queries is not understood
Could you explain more equation 1, table 1 and figure 14. I have changed it. I have struggled with choosing a distribution because there is no facet on this matter. I have concluded (as also other experts) that the distribution is not known outside the range of query (Bernoulli/Binomial 0 1). I have changed the result section and made new analysis. Hopefully, the section has gotten better and provide a context for present work. Since EA is a condition that appear frequently it is not a rare event, but I compare it to a rare event. We need more research to be able to define the final distribution for EA also with respect to present ontology. The ontology produces 0 and 1. Your comment made me write what I really man about this question. It is a question that should have its own project. Even mathematicians at high level I hava talked with agree that we do not know.
-In 2.3 you should mention that each query is a model that is just observed in appendix 2. Query examples Yes, I see the point. Have made it explicit.
-It does not place the OFEDR ontology or any of its parts as classes, subclasses and properties
What software did you use to build the ontology? I used Protégé and that is referred to in the introduction (aprox line 200), method section (under instruments were also eye tracking is added), abstract and in acknowledgments. I have also included Weka in the method section below instruments. I have made a description of how it is possible use ontology connected directly to machine learning, but the point with this work is to show that the ontology can be used on its own. Results from queries can also be used to produce new data samples.
You should specify more about the ontology metrics in the text of the article only mention a table in appendix 3 I have tried to do that and moved the table into the result section.
-In the conclusions section you mention that the literature review is a product of the results of the work but it is part of the literature search. Yes, it was an error. I have changed it. And changed conclusion du to change in result section related also to our comment about distribution.
It is the first time I try this system by sending reply and documents. Hope it works.
Thank you very much for your effort.
Best regared,
Linda Wendelberg

Reviewer 2 Report
The author presents an interesting ontology grounded on an eloquent philosophical perspective. A technical implementation of the ontology is described. However, there are some weaknesses that must be addressed:
- The assumption of a Poisson distribution to detect an extreme condition is not justified. In fact, from complex systems theory, it is well known there are better probabilistic models to detect extreme events.
- The threshold to qualify a radicalization process is set to four objects but no rationale is provided to support this choice.
- Even though a general definition of radicalization or terrorism is not available, at least some typical features about radicalized or terrorist behavior should be provided. This is convenient in order to draw a clearer picture about what it is implied in those terms.
Author Response
Dear Reviewer,
Thank you for your work with reviewing the manuscript. I reply to each of your comments step by step.
First, English proofing on the entire manuscript is done.
- The assumption of a Poisson distribution to detect an extreme condition is not justified. In fact, from complex systems theory, it is well known there are better probabilistic models to detect extreme events. I have had many rounds on your comment of this aspect also before I sent the article. EA is not an extreme rare event. This is actually what is so interesting with present finding. Because even if all in the experiment group is in the condition the probability of detecting them based on risk is low. On the other hand, terrorism is a very rare event. I have changed the whole result section and tried to combine the raw measures of probabilities from models with the terrorism probability to visualize this point in the article. I suggest that detection of radicalization is not detection of a rare event. Many people are radicalized without executing terrorist attacks. Maybe something prevented that the radicalization escalated further. E.g. because something changed in their life path (e.g., that someone saw the change made an intervention). However, your comment contributed to increased meaning for this work because as can be seen there is a gap. The gap is between possible early signs of radicalization and the risk of terrorism. It is inside this gap the preventive work must be done. I contacted some mathematicians to hear their opinion and they meant that we do not know the distribution. However, I think this is a very interesting subject, which could be an interesting project.
- The threshold to qualify a radicalization process is set to four objects but no rationale is provided to support this choice. See your point. I forgot to write about it. It is based on machine learning testing. I have included some of those results in the result section. It is possible to develop both good and bad models based on different numbers of variables, but no good models had numbers of variables below 4. Especially does knowledge about personality increase the accuracy. I have tried to explain that in the manuscript in the result section. I hope I reply to what you meant.
- Even though a general definition of radicalization or terrorism is not available, at least some typical features about radicalized or terrorist behavior should be provided. This is convenient in order to draw a clearer picture about what it is implied in those terms. Correct there is something (and lot of meanings but not much emperi) and I have reorganized the introduction and included more information about “state-of-the-art” for terrorism. Of especially importance for my work is the potential relationship between extremism and sexual change (fetish stimuli is used in data gathering). At moment there are many projects working on treatment policies and treatment intervention related to radicalization. It is not impossible that this in future may develop being a diagnosis. Time will show. Small pupil diameter is mentioned a couple of places in the manuscript. This is very interesting finding combined by the knowledge about EA because EA is related to lower insula activation. In this region we find some processing related to pupil diameter (in addition to other sites in the brain). I have referred to some additional research and made a broader description in the introduction. I think that there exist a lot of knowledge but it is not available for the public.
I hope I have managed to reply to your comments and have interpreted the comments correctly. There is knowledge about terrorist traits in the world but I think the people who works with this cannot say much. It similar in medicine and psychology something is not told openly. But that they have less feelings (see also descriptions in section 3.0.4 (table 5) make some sense.
Thank you very much for your effort in reviewing the manuscript.
Best regards,
Linda Wendelberg

Reviewer 3 Report
Review: An Ontological Framework to Facilitate Early Detection of ’Radicalization’ (OFEDR) –A Three World Perspective
Verdict:
I am in favour of the publication after small revisions
This is a very interesting piece that provides very good ideas on radicalisation. The author demonstrates very good theoretical and empirical knowledge on the area, which is a good foundation for any future publication. Nonetheless, I believe there are some small, minor issues that could be addressed. Therefore, I recommend an acceptance in principle which would require the author to make some small changes to the article in the light of my suggestions below.
Firstly, the paper is coherently and clearly written. The author provides a clear framework for analysis, which is articulated, explained and put in its context within the literature of the article. The author does a good job at outlining some of the main authors to the debate, their contributions and the insights taken from them which are applied in the article. The author provides a good number of important sources and secondary literature in the field. On the whole, the article is well balanced and makes reasonably argued points. I can agree on the larger points the author is making, although I might have some minor quibbles.
Some general points for changes:
- The introduction in its current state could be snappier and clearer: why is this topic interesting? I know the topic is interesting, but this could be underlined more strongly.
- The conclusion of the article should discuss how the findings of the article could be generalised. Can the findings be generalised, if yes, why? If no, why not? This is an important task for the author.
- What is the methodology of the article? The justification for case selection?
- What do we learn from the article?
Nonetheless, the required changes should not detract from the fact that this is a very clearly argued and articulated article, with a generally convincing thesis, which warrants publication after the revisions have been made.
Author Response
Dear Reviewer,
Thank you for reviewing the manuscript. I have tried to make changed as many as I could on less than five days. I will reply to your comments step by step. I am happy that you liked the subject because few understand what I am working on in my projects. This condition is related to develop ment of antisocial "mind" and I speculate whether it can be related to development of psychiatric diagnosis. We do not know that today. Now your comments.
1. The introduction in its current state could be snappier and clearer: why is this topic interesting? I know the topic is interesting, but this could be underlined more strongly.
I have tried to solve that by including more information about terrorism concrete as also suggested by another reviewer. Moreover, the research hypotheses (H1) are stated clearer and changed. This may also relate to result section, which is also changed. The numbers are not changed but how they are used are changed. So, I hope that the change in introduction connect to the result section and the discussion.
2. The conclusion of the article should discuss how the findings of the article could be generalized. Can the findings be generalized, if yes, why? If no, why not? This is an important task for the author. I changed the heading of subsection 3.03 and called it generalizability. Last in this section it is summarized what can be generalized based separate variables in the models both for simulated values and test data. It is also added more information about the sample for testing in the method section. I think generalizability can be done but it depends on type of data. For example, does subjects in this condition respond different to novel and familiar stimuli. But if this knowledge is used it can be generalized. Brain function also tell a lot even if we cannot see it. Some information about brain can be inferred from eye tracking data. I used table because it is so much information in the article. Hope that is ok.
3. What is the methodology of the article? The justification for case selection? Good point I had forgot to write about it. I have tested different models and some results from this testing are now included in the result section. These tests are based on my test data and not on findings from other researchers (simulated values). I have also included more information about the test sample in method section. For example, I used reduced body awareness because there is good documentation on that aspect. And these findings are again supported by findings about how brain activates in this condition. So, it is very complex. I hope I have included enough information without it becoming too much.
4. I suggest that when you refer to case selection it is why I have made those five models. I have made one explanation about choosing test variables in the models in the result section. I had forgot to write about this and was uncertain whether I should. I have added more information about test sample in method section. In supplementary section and in new section 3.0.4. there is information about other scientific evidence. The result section is change hopefully to the better. The results are not changed but how they are presented and analysis. Simplicity is frequently a good think. Raw probabilities are ease for all of use to intuitively understand. See especially new comments in Table A2 in supplementary section.
5. What do we learn from the article? I am inside the matters and think others sees it. However, it is a very good point to make this aspect more concrete and be clearer about hypothesis. After changes made in result, method, introduction and abstract I hope it is clearer. Detection by using estimates for terrorism cannot be used in detection of early signs of radicalization. Extreme probability values do not contribute to detection of preventive value. It also shows that existing knowledge about radicalization and EA can be used to develop models for early detection of radicalization (“the ontological system”). I hope it is clearer. The reviews helped to make new thinking as well as making me secure enough to write what I really see.
English proofing on the entire manuscript is done.
Thank you very much for your effort with reviewing the manuscript. I hope it is better.
Best regard,
Linda Wendelberg
